# Glaucoma Animal Models beyond Chronic IOP Increase

**DOI:** 10.3390/ijms25020906

**Published:** 2024-01-11

**Authors:** Teresa Tsai, Sabrina Reinehr, Leonie Deppe, Alexandra Strubbe, Nils Kluge, H. Burkhard Dick, Stephanie C. Joachim

**Affiliations:** Experimental Eye Research Institute, University Eye Hospital, Ruhr-University Bochum, In der Schornau 23-25, 44892 Bochum, Germany; teresa.tsai@rub.de (T.T.); sabrina.reinehr@rub.de (S.R.); leonie.deppe@rub.de (L.D.); nils.kluge@rub.de (N.K.); burkhard.dick@kk-bochum.de (H.B.D.)

**Keywords:** glaucoma, normal-tension glaucoma, animal model, ischemia–reperfusion, autoimmune glaucoma, excitotoxicity, nerve crush

## Abstract

Glaucoma is a complex and multifactorial disease defined as the loss of retinal ganglion cells (RGCs) and their axons. Besides an elevated intraocular pressure (IOP), other mechanisms play a pivotal role in glaucoma onset and progression. For example, it is known that excitotoxicity, immunological alterations, ischemia, and oxidative stress contribute to the neurodegeneration in glaucoma disease. To study these effects and to discover novel therapeutic approaches, appropriate animal models are needed. In this review, we focus on various glaucoma animal models beyond an elevated IOP. We introduce genetically modified mice, e.g., the optineurin E50K knock-in or the glutamate aspartate transporter (GLAST)-deficient mouse. Excitotoxicity can be mimicked by injecting the glutamate analogue N-methyl-D-aspartate intravitreally, which leads to rapid RGC degeneration. To explore the contribution of the immune system, the experimental autoimmune glaucoma model can serve as a useful tool. Here, immunization with antigens led to glaucoma-like damage. The ischemic mechanism can be mimicked by inducing a high IOP for a certain amount of time in rodents, followed by reperfusion. Thereby, damage to the retina and the optic nerve occurs rapidly after ischemia/reperfusion. Lastly, we discuss the importance of optic nerve crush models as model systems for normal-tension glaucoma. In summary, various glaucoma models beyond IOP increase can be utilized.

## 1. Introduction

### 1.1. Prevalence of Glaucoma

Glaucoma is one of the most common causes of blindness worldwide [1]. According to the National Eye Institute, about 2.7 million people in the United States are currently suffering from glaucoma. This number is expected to rise to 4.2 million in 2030 (https://www.nei.nih.gov/learn-about-eye-health/eye-conditions-and-diseases/glaucoma; accessed on 13 December 2023). There is a rising trend due to the aging of society. This causes an immense socio-economic impact and leads to enormous impairments of patients’ independence and quality of life.

### 1.2. Pathophysiology of Glaucoma

Glaucoma as a group of eye disorders is characterized by progressive optic nerve damage and the loss of retinal ganglion cells (RGCs) [2,3]. This results in visual field loss, which ultimately leads to blindness. There are primary as well as secondary forms of glaucoma. Primary open-angle glaucoma is defined as a chronic, progressive, potentially blinding disease, causing optic nerve rim and retinal nerve fiber loss with related visual field defects. Major risk factors include elevated intraocular pressure (IOP) as well as advanced age [2]. Normal-tension glaucoma (NTG) involves optic nerve damage and RGC loss even when the IOP remains within the normal range [4,5]. In secondary forms of glaucoma, another eye condition or health issue such as injury, inflammation, or certain medications is causing this disease.

### 1.3. Glaucoma Treatment Options

Early stages of glaucoma may not cause noticeable symptoms, which is why it is often not diagnosed until critical damage has already occurred. While there is no cure for glaucoma, it can usually be managed with IOP-lowering treatments. Eye drops are often the first choice of treatment, thereby reducing the production of aqueous humor or improving its drainage [6,7,8]. However, this approach is not always successful. Also, compliance is crucial for the effectiveness of this medication. The application of glaucoma medication as eye drops can lead to side effects such as eye irritation, blurred vision, or bradycardia, which reduce patient compliance and accordingly limit the success of the therapy [9]. The next step is often laser therapy, such as trabeculoplasty, iridotomy, or cyclophotocoagulation, or surgery. Different surgeries can be carried out to lower IOP [10]. Minimally invasive glaucoma surgery is quite novel and on the rise [11,12,13]. Glaucoma drainage devices or trabeculectomy are also an option. Nevertheless, early detection and proper management are essential for preserving vision and preventing further damage to the optic nerve [14].

Currently, only the increased IOP, as a main risk factor, is used as the treatment target [15]. This can slow down the progression of the disease but not cure it. Also, this therapy does not work for all patients, and not all forms of glaucoma involve elevated IOP, like NTG. The goal of glaucoma care should be to promote the well-being of patients and their quality of life [2], which are influenced by a person’s visual function and the psychological impact of having a chronic, progressive, and sight-threatening disease [2]. It is therefore necessary to develop new treatment options and ultimately improve treatment success rates.

The exact etiology of this disease remains unclear. Excitotoxicity, immunological mechanisms [16,17] including an inflammatory microglia response [18,19], ischemic processes [20,21], and oxidative stress [22,23] are thought to be involved. Evidence in favor of an immunological involvement in the development of glaucoma includes the altered autoantibody patterns found in glaucoma patients [17,24].

### 1.4. Glaucoma Animal Models

Animal models are commonly used in research to better understand diseases, explore their underlying mechanisms, and test potential treatments. Various animal species have been employed to simulate disease-like conditions and study their effects. To understand the complex glaucoma pathology and to analyze novel treatments, different glaucoma models are currently employed [25,26]. As in other fields of medical research, rodents are mainly used for in vivo studies. But rabbits [27], dogs [28], or non-human primates [29] are also utilized in glaucoma research. Porcine models have gained interest in recent years due to their larger eyes and similarities in eye anatomy to humans [30]. Zebrafish are used to study the genetic and molecular aspects [31]. Many of these models, especially in mammals, are based on a chronic IOP elevation and called ocular hypertension (OHT) models. Here, researchers artificially induce elevated IOP in animals by blocking the drainage pathways, injecting substances like hypertonic saline, or creating a genetic alteration that blocks aqueous humor outflow [32,33,34,35,36]. As previously mentioned, glaucoma is a complex and multifactorial disease, and elevated IOP is not the only pathogenic factor. Hence, in addition to OHT models, there is a need for models where aspects of glaucoma that are not solely dependent on high IOP can be investigated. For example, other contributing factors like genetic mutations, vascular changes, and neurodegenerative processes need to be considered. This review aims to provide an overview of glaucoma animal models beyond IOP elevation. It will focus on genetic or immune-based NTG models. It will also cover different toxic and ischemic as well as nerve crush models (Figure 1).

## 2. Genetic Modulation

In glaucoma, as in many other neurodegenerative diseases, genetic changes play a crucial role. In addition to many other biomarkers, several genetic alterations have already been identified in the course of diagnostic procedures in glaucoma pathology. Therefore, a way to further examine glaucoma, especially NTG, is to induce genetic dispositions that can reflect the identified abnormalities in patients [34].

### 2.1. Genetically Modified Normal-Tension Glaucoma Zebrafish Model

Although many genes are associated with cases of glaucoma in humans, incomplete penetrance within families suggests that the various causes of glaucoma are multigenic in nature [37]. The zebrafish, in turn, is an ideal model system for studying multigenic traits [38,39]. In humans, an association of the forkhead box C1 transcription factor (*FOXC1*) gene with Axenfeld–Reiger syndrome has been identified, which also includes glaucoma in some individuals [40]. Reduced *foxc1* expression during zebrafish eye development also results in reduced expression of the cellular homeostasis and apoptosis regulator forkhead box O1 (*foxo1*) and increased apoptosis in the zebrafish eye [41]. Moreover, mutations in this transcription factor cause glaucoma-like pathologies, such as abnormalities in the iris, trabecular network, and cornea [31,42]. A mutation of the LIM homeobox transcription factor 1-beta (*LMX1B*) gene in humans was also associated with increased susceptibility to glaucoma [43]. By reducing the *lmx1b* expression in zebrafish with morpholinos, early morphological defects in the eye and increased fibroblast growth factor activity in the eye could be detected. Moreover, an altered expression of two other genes that are also involved in glaucoma, the already-mentioned *foxc1* and the paired-like homeodomain transcription factor 2 (*pitx2*) gene, could be observed [44,45]. The results of the study also suggest that *lmx1b*-expressing cells in the zebrafish eye exhibit migration defects in the absence of *lmx1b* expression [45].

### 2.2. Genetic Modified Rodent Normal-Tension Glaucoma Models

In 2002, studies highlighted a significant correlation between a mutation in the gene for optineurin (OPTN) and glaucoma development in patients [46,47]. Optineurin is a cytosolic protein which is involved in many cell processes like vesicle trafficking, the nuclear factor kappa B pathway, cell division, and autophagy. Its mutation is also linked to other diseases such as, for example, amyotrophic lateral sclerosis [48]. The *OPTN* E50K mutation was found in hereditary NTG patients, and it is also the most frequently occurring mutation in these cases [47,49]. Based on these findings, one of the first genetically modified rodent animal models was established, the optineurin E50K knock-in mouse (E50K; Figure 1) [50]. Even if the precise functionality of this gene is not quite revealed yet, it is associated with the tumor necrosis factor alpha (TNF-α) signaling pathway, and several physiological processes, like mentioned before, the secretory vesicle transport and autophagy [47,51,52]. Optineurin is expressed in different parts of the eye. The induced overexpression in E50K mice leads to glaucomatous-like damage without increased IOP, evidenced by an increased apoptosis rate leading to RGC loss and a reduction of the retinal thickness [53,54]. An additional study indicated a correlation between specific miRNAs and the genetic modification of optineurin. They used the E50K mice to investigate the expression patterns and found significant alterations, indicating a pathological connection [55]. However, it is still unclear how this specific degeneration is caused. Although this model can mimic the genetic mutation of NTG patients and is therefore a suitable model, it must be taken into account that patients with this gene mutation only represent approximately 10% of cases worldwide [56].

Myocilin is also a protein that became known in connection with research regarding glaucoma. Approximately 5% of primary open-angle glaucoma cases in patients result from a mutation in the gene for myocilin, and it is expressed on higher levels in the trabecular meshwork [57,58]. Here, it has to be mentioned that this mutation alters the IOP and therefore does not serve as a specific marker for IOP-independent glaucoma pathology [59]. Nevertheless, suitable mouse models have already been developed in this direction, in which new findings about the described genetic disposition, the changes in the endoplasmic reticulum stress level, and the associated glaucoma pathology could be discovered [60].

Furthermore, it could be shown that the expression of certain glutamate receptors/transporters is reduced in human glaucoma patients. Glutamate is the most important and prominent excitatory neurotransmitter in the central nervous system (CNS) and, consequently, also in the retina. It is generally deduced that increased extracellular glutamate, which can no longer be absorbed, is the cause of excitotoxicity and the following glaucomatous damage in NTG patients [61,62]. Another genetic and IOP-independent model is based on this excitotoxicity of extracellular glutamate and the fact that severe exposure to the excitatory glutamate leads to neuronal cell death (Figure 1) [63,64,65]. The absence of specific glutamate transporters therefore leads to RGC death and further glaucoma-like damage. This fact was used to establish glutamate aspartate transporter (GLAST)-deficient mice, in which mutations in the solute carrier family 1 member 3 (*Slc1a3)* gene lead to a lack of glutamate/aspartate transporters in Müller cells, which in turn increases the extracellular glutamate level [66,67]. In these mice, cupping of the optic nerve and degenerated axons could be observed [68,69,70]. In this case, it is interesting that the ingested glutamate is subsequently converted into glutathione, an endogenous non-enzymatic antioxidant. The synthesis takes place in the cytosol and depends on the amino acid cysteine, glutamate, and glycine. The reaction is adenosine triphosphate-dependent and is carried out by the glutamate-cysteine ligase and the glutathione synthetase. Its antioxidative functions result from its capability to scavenge free radicals and to work as a cofactor for the enzymes glutathione S-transferases and peroxidases [71]. Since oxidative stress plays a key role in the development of glaucoma, the genetic deficiency of glutamate transporters and the resulting lack of glutathione could lead to increased oxidative stress. To conclude, the mutation also influences this part of the development of glaucoma.

Similar results were obtained with the excitatory amino acid carrier 1 (EAAC1) mouse model, which is characterized by the absence of the excitatory amino acid carrier 1, another important glutamate transporter located in RGCs. Here, changes could already be detected after eight weeks, much earlier than in the GLAST model with eight months [69,72]. In previous studies with both mice lines, it was possible to reduce the glaucomatous damage via glutamate receptor agonists or oxidative stress level reduction [73,74,75]. Both mouse lines exhibit a normal IOP, which makes both glutamate-dependent knock-out mouse models a promising and successful approach for conducting analyses in the field of NTG research (Figure 1). Naturally, even these approaches also have their limitations, such as the fact that degeneration is not focused on a specific area of the retina as in glaucoma patients but rather affects the entire tissue.

## 3. Excitotoxicity

Besides other factors, an overstimulation of the excitatory amino acid glutamate plays a role in the degenerative processes in glaucomatous damage. Glutamate is the main stimulatory neurotransmitter of the CNS and is a nonessential amino acid [76]. Production occurs through endogenous mechanisms and not through the ingestion of food [77]. It is important in a variety of processes, for example, in amino acid metabolism or energy balance [77]. In addition, glutamate also performs neurobiological tasks by regulating motor as well as brain functions [78,79]. However, glutamate overstimulation can also cause neurotoxic processes, which are also observed in some neurodegenerative diseases, including Alzheimer’s and Parkinson’s disease [80,81,82,83].

### 3.1. Excitotoxicity-Based In Vivo Glaucoma Models

When potentially toxic substances are injected intraocularly, the substance induces cell death, which activates further molecular processes that either stop or enhance cell death. Among these toxic substances, as already mentioned, is glutamate or the glutamate analogue N-methyl-D-aspartate (NMDA). Thus, it is possible to study the effect of these substances, which have been found to be damaging factors in neurodegenerative diseases, in different model systems (Figure 1). As with the genetic models, there are zebrafish ones in which NMDA induces retinal degeneration similar to glaucoma damage [84,85]. The intravenous injection of NMDA is best used for the analysis of acute retinal damage, while the intramuscular administration of NMDA seems to be better-suited for the analysis of chronic retinal damage [86]. Moreover, it is known that an intravitreal injection of NMDA in rats and mice leads to a rapid degeneration of RGCs starting after just one day [87,88]. Honda et al. distinguished between two sub-types of RGCs after intravitreal NMDA injection in C57/Bl6 mice. Seven days after the injection, they noted a loss of about 60% of RNA-binding protein with multiple splicing (RBPMS)^+^ labeled RGCs. However, neither the number of α-RGCs nor of intrinsically photosensitive RGCs was altered, suggesting that they are highly tolerant to NMDA-induced damage [89]. Furthermore, it was shown that after NMDA injection, higher levels of oxidative stress were detectable [90,91]. Conclusively, Maekawa et al. found that the oral administration of plant-derived antioxidant compounds could protect against RGC degeneration after NMDA injury [92]. Also, coenzyme Q10 and alpha-tocopherol reduced the retinal damage and the number of apoptotic cells in this model [93]. In a recent study, NMDA-induced damage could be abolished by treatment with an adenosine A1 receptor antagonist [94]. However, not only the RGCs in the retina but also the optic nerve is affected by excitotoxicity. For example, axonal degeneration could be observed by toluidine blue staining seven days after NMDA injection [95,96]. By examining different points in time, Kuehn et al. revealed that degeneration occurred later in the optic nerves than in the retina. This led to the suggestion that the damage is mainly a consequence of the cell body loss in the retina and not caused by the additive excitotoxic effects of NMDA [87]. A study by Yan et al. investigated the hypothesis that depending on the location of NMDA receptors, synaptic or extrasynaptic, cell death or survival is promoted. They observed that a complex of the NMDA receptor and the transient receptor potential cation channel subfamily M member 4, which is formed extrasynaptically, is required for excitotoxicity [97].

### 3.2. Excitotoxicity-Based Organ Culture Models

Ex vivo retinal organ cultures can be a useful tool to study glaucoma-like damage (Figure 1). The porcine eyes offer the advantage that they are more similar to human eyes than those of rodents. Studies showed that porcine explants can be in culture for about 8 days [98,99,100]. Hence, NMDA damage can not only be investigated by using in vivo animal models. In the ex vivo porcine organ culture model, NMDA was supplemented at three different concentrations. After 7 days, NMDA treatment significantly increased the apoptosis of RGCs in all the NMDA groups, without causing a profound RGC loss. Moreover, NMDA induced oxidative stress as well as mild microglia activation [101]. In addition to NMDA, other excitotoxicity-based substances have already been successfully established in this easily reproducible and cost-effective porcine organ culture model to mimic various aspects of glaucomatous damage [102]. Appling hydrogen peroxide (H_2_O_2_) to the organ culture also induced oxidative stress, which led to severe degeneration of the RGCs with persistent apoptosis and a strong microglial reaction [100]. In order to imitate the factor of hypoxic stress, which can also be observed in glaucoma patients, cobalt chloride was used in the porcine organ culture. It was shown that cobalt chloride application not only damaged the RGCs in the retina, but also the amacrine and bipolar cells. Therefore, in this model system, the deeper layers of the retina are also affected [99]. In the meantime, various potential substances have been examined for their protective abilities in both the H_2_O_2_ and cobalt chloride damage model. It was shown that the inducible nitric oxide synthase (iNOS) inhibitor 1400W can protect against both oxidative and hypoxic damage [100]. Interestingly, extremolytes such as ectoine and hydroxyectoine also had a protective effect on the hypoxia-damaged retina in the porcine organ culture model and thus represent a potentially interesting therapeutic approach for glaucoma patients [103]. Another study examined whether application of glial fibrillary acidic protein or gamma-synuclein can protect RGCs in the porcine organ culture model. The explants were incubated for 24 h and evaluated afterwards. The results showed a protective effect of both substances regarding the number of RGCs [104]. Commonly, intravitreal injections are used for the application of medications to the eyes. Many natural barriers, like the vitreous body, complicate or even prevent the reaching of the drug to the retina. Here, the ex vivo model can be utilized to improve or investigate possible enhancements, including nanoparticles, for the medications to reach the retina [105,106].

## 4. Immune-Based Glaucoma Models

### 4.1. Contribution of the Immune System in Glaucoma

Several studies have indicated the involvement of the immune system in the pathogenesis of glaucoma. Increased autoantibody titers against ocular tissue antigens were observed in the sera and aqueous humor of normal- as well as high-pressure glaucoma patients [107,108,109,110]. Also, proteins of the immune system, including the complement proteins C1q, C3, and C8, were found to be upregulated in the aqueous humor of glaucoma patients. Moreover, higher levels of markers for macrophages and neutrophils were also identified [111,112]. Glaucoma patients also showed an increased infiltration of heat shock protein (HSP)-specific T-cells. Additionally, in an OHT model, IOP elevation led to RGC loss and optic nerve degeneration. Moreover, infiltrations of autoreactive T-cells into the retina were noted. These T-cells were pre-sensitized by the commensal microflora. Moreover, an upregulation of extracellular and membrane-bound HSP27 in the ganglion cell layer was observed. Hence, it was discussed whether HSP27-imprinted CD4^+^ T-cells penetrate the retina and thus cross-reacted with the HSP27 expressing cells of the ganglion cell layer, which led to a loss of RGCs in an animal model [113]. The pathophysiology of glaucomatous damage seems to involve T-cell-mediated processes [113,114].

### 4.2. The Experimental Autoimmune Glaucoma Model

To investigate to what extent the immune system is involved in the pathogenesis of glaucoma, Wax et al. developed an IOP-independent in vivo rat model through immunization with HSPs [115]. Later, it was shown that the loss of RGCs as well as the degeneration of the optic nerve can be observed after immunization with different antigen homogenates like the optic nerve antigen (ONA) [116] or the retinal ganglion cell layer homogenate [117]. The first studies in this so-called experimental autoimmune glaucoma (EAG; Figure 1) model indicated that the loss of RGCs is facilitated through caspase-mediated apoptosis [118]. Furthermore, Joachim et al. observed immunoglobulin G (IgG) deposits in the retinal ganglion cell layer, in close vicinity to apoptotic cells [117]. This raises the question of whether antibodies are a cause or consequence of glaucoma. It is known that IgG autoantibodies can activate the complement system via the classical pathway [119]. Interestingly, activation of the complement system in the EAG model was mainly observed via the lectin pathway at the examined points in time [120]. Although a significant RGC loss was not detected until 28 days after immunization, increased activity of the complement system was noted after 7 days. This indicates that the dysregulation of the complement cascade is closely linked to cell death and possibly even triggers it. The number of microglia/macrophages also seems to increase after immunization [117,121]. Microglia are the macrophages of the CNS, yet it is known that microglia also have the ability of cellular cytotoxicity [122]. Recent proteomic data further confirmed that the immune system plays an important role in the pathology of glaucoma since a shift in the regulation was observed in several immune system-associated proteins after immunization with S100 calcium-binding protein B (S100B), including HSP60 [123]. Altogether, the immune system seems to be a promising target for new therapeutic approaches. In the first studies, intravitreal injection of an antibody against the complement factor C5 resulted in the prevention of RGC and retinal function loss by reducing the formation of the terminal membrane attack complex and diminishing inflammatory as well as microglia cells [124]. Immune regulation can be associated with proteins of the extracellular matrix (ECM). In an OHT model, the ECM protein tenascin-c as well as several immune system markers were found to be upregulated [125]. However, not only in an IOP-dependent model, but also in EAG rats, an increase in tenascin-c and phosphacan was revealed in retinae and optic nerves after ONA or S100B immunization. This remodeling occurred especially before glaucomatous damage was noted [126]. Traditionally, the EAG model was established in rats; to further facilitate this model, Reinehr et al. successfully transferred it into mice [121].

### 4.3. Combination of the EAG Model with IOP-Dependent and -Independent Glaucoma Models

Animal models in mice have the advantage of specifically knocking out or knocking in genes that can be used to identify the mechanisms underlying glaucoma more precisely. This advantage was used to immunize tenascin-c knock-out mice. Ten weeks after immunization, Wiemann et al. could show that in tenascin-c knock-out mice, the glaucomatous damage was less severe compared to that in wildtype EAG mice. Moreover, the inflammatory response was diminished in tenascin-c knock-out mice which were immunized with ONA in comparison to wildtype animals that also received ONA [127]. The transfer of the EAG model from rats to mice was further used to establish a new combinatorial model based on the EAG model and the transgenic high-pressure connective tissue growth factor model to better represent the multifactorial component of the disease. The first studies in this novel multifactorial model revealed an additive degeneration of both optic nerves and RGCs. This was accompanied by an increase in microglia/macrophages in the retinae [128]. In summary, there is no doubt that the immune system is an important risk factor in the pathogenesis of glaucoma. Further studies in the EAG model as well as in the multifactorial model may help to deepen the understanding of that factor, which could lead to new therapeutic approaches for glaucoma.

## 5. Ischemia Models of Glaucoma

Retinal ischemia is defined as a lack of blood supply to the retina which is related to an undersupply of oxygen and nutrients [129]. The hypoxia leads to the activation of hypoxia-inducible factors, which mediate neovascularization via vascular endothelial growth factors. Concurrently, the failure of energy metabolism causes excitotoxicity. Microglia affected by ischemia contribute to inflammation by secreting proinflammatory cytokines. In addition, the blood-retinal barrier is damaged, resulting in infiltration of the retina by leukocytes and macrophages. These increase inflammation and vascular dysfunction [130,131]. Ischemia is followed by reperfusion, which is the restoration of the blood supply [129]. Reperfusion is accompanied by increased oxidative stress caused by the accumulation of reactive oxygen species [132]. In summary, ischemia and reperfusion cause severe neuronal damage within the retina and optic nerve, leading to impaired retinal function [133,134,135]. Ischemia contributes to the pathophysiology of various retinal diseases, including glaucoma. Different animal models have been developed which enable the investigation of structural and functional changes after retinal ischemia as well as the testing of therapeutic approaches [136]. Most studies use rodents because of the lower organizational effort and cost [137]. A commonly utilized model is the ischemia/reperfusion (I/R) model, in which a transient increase of the IOP leads to an impaired retinal blood flow. Alternatively, the central retinal artery can be occluded by clamping or photothrombosis, or vasoconstriction of retinal vessels can be initiated by the intravitreal application of endothelin [136].

The I/R model is a frequently used glaucoma model (Figure 1) [138,139,140,141]. A thin needle connected to a raised saline reservoir is used to puncture the anterior chamber of the eye. Hydrostatic pressure is increased above systolic blood pressure and causes retinal ischemia by compressing the retinal blood vessels. Lowering the reservoir starts the reperfusion. The duration of ischemia as well as the level of IOP vary depending on the model design and research question. The I/R model allows us to examine the effects of acute and transient retinal ischemia. Another advantage of this set-up is its reproducibility. Disadvantages include the need for surgical skills and prolonged anesthesia [137]. In addition, the increased pressure itself may cause damage to the retina [136].

The I/R injury affects the whole retina, with damage increasing over time. The impairment includes a decrease in the thickness of the entire retina. The loss of RGCs starts within 2 h and shows a progressive course [133,135]. In addition, the loss of cholinergic amacrine cells, glycerinergic AII amacrine cells, cone bipolar cells, and cone photoreceptor cells can be detected. Rod bipolar cells and rod photoreceptor cells seem to be more resistant to ischemic damage [138]. The underlying cell death mechanism is, specifically, apoptosis [138]. Cell loss is reflected in impaired retinal function, which can be measured by electroretinography (ERG) measurements. Amplitudes of the a- and b-waves are already reduced 3 days after ischemia and show a progressive decrease over the course of time [133,135]. Beside the neuronal cells of the retina, glial cells, both macroglia and microglia, are affected by ischemia [142,143,144,145]. The activation of microglial Toll-like receptors (TLRs), including TLR2, TLR3, and TLR4, leads to the release of proinflammatory cytokines, such as interleukin (IL)-1β, IL-6, TNF-α, and transforming growth factor beta, which amplifies the inflammatory environment and enhances neuronal cell death [146]. In addition to the retina, there is also progressive damage to the optic nerve, which appears in the form of tissue dissolution, demyelination, cell infiltration, and gliosis [20,134,147]. Furthermore, the composition of the ECM of the retina as well as that of the optic nerve changes after ischemia. This includes altered expressions of the glycoproteins tenascin-C, fibronectin, and α1-laminin, which contribute to ischemia-induced damage [148,149].

The I/R model is commonly used for testing novel therapeutic approaches [21,150,151,152]. 

## 6. Optic Nerve Crush or Transection Models

Since the nerve fibers of the optic nerve can show an early sign of injury in glaucoma, optic nerve injury represents certain characteristics of optic neuropathy seen in this disease. Optic nerve crush and partial or complete optic nerve transection in rodents are commonly used models [153,154,155,156]. A lesion to the optic nerve leads to the retrograde degeneration and, finally, to the death of RGCs. The time courses of RGC loss after optic nerve crush and transection are comparable. A total of 65% of RGCs die within the first seven days [157]. Moreover, a moderate decrease in the dendritic structure in the remaining RGCs can be observed [158]. The structural damage to the retina is accompanied by functional impairment. The amplitudes of the ERG pattern are significantly reduced three days after optic nerve crush and eliminated after seven days [159]. The cell death mechanism of RGCs is primarily apoptosis [157]. Shortly after optic nerve transection, proapoptotic (*Ei24*, *Gadd45a*) as well as anti-apoptotic genes (*Iap-1*) are simultaneously upregulated [160]. Using the nerve crush model, numerous therapeutic interventions have already been tested for their effectiveness. Intraperitoneal minocycline treatment in the case of optic nerve transection showed anti-apoptotic properties [161]. In addition, intravitreal treatment with human NgR1(310)-Fc increased RGCs with regenerating axons after optic nerve crush [162]. Moreover, several laboratories routinely use intravitreal injections of adeno-associated virus serotype 2 (AAV2) in the optic nerve crush model to investigate possible mechanisms that gene delivery has on the neuroprotection and axon regeneration of RGCs [163]. Among other things, it was shown that the activation of the Sigma-1 receptor in the retina by AAV in the optic nerve crush model led to the protection of both the retinal structure and its functionality [164]. In another study, the kinetics of the bcl-2-associated x protein (BAX) activation using a AVV2-mediated BAX-GFP gene transfer were compared between the 611 W cell culture and the optic nerve crush model. BAX is a component of the RGC death signaling pathway and, therefore, an attractive target for future neuroprotective therapy options. The comparison between the BAX activation kinetics of the cell culture and the optic nerve crush model shows clear differences and underlines the importance of animal models for glaucoma and therapy research [165].

## 7. Marmosets with Naturally Occurring NTG

*Callithrix jacchus*, the marmoset, a small New World non-human primate, is a suitable model for neuroscience research. They resemble humans with a high similarity and have a short lifespan. A study by Noro et al. investigated whether aged marmosets develop glaucoma. The results yielded that 11% of these animals showed characteristics of glaucoma disease. Without an elevated IOP, an increase in oxidative stress as well as lower levels of the brain-derived neurotrophic factor and its receptor, tropomyosin receptor kinase B, was noted in the retina. This was accompanied by a thinning of the retinal nerve fiber layer and an atrophy of the primary visual cortex. Hence, the authors conclude that the marmoset can be a potential model to study NTG more precisely [166,167].

## 8. Conclusions

The exact mechanisms that lead to glaucoma, one of the most common causes of blindness, are still unknown. In addition to increased IOP, other mechanisms play a crucial role in the development and progression of glaucoma. Excitotoxicity, immunological changes, ischemia, and/or oxidative stress are known to contribute to neurodegeneration in glaucoma disease. There are now numerous different animal models that reflect these factors and therefore mimic different pathogeneses of glaucoma. Each of the models mentioned has its own strengths and weaknesses (Table 1). Some of these models have been used successfully by several researchers with consistent results, whereas others have only been known for a short time and need to be independently reproduced in other laboratories. Given the advances in better understanding the molecular pathogenesis of glaucoma in humans and new molecular techniques to selectively alter the expression of specific genes in certain tissues, new opportunities are also emerging for the development of additional animal models of glaucoma.

The development of new glaucoma models, both using novel techniques and through the combination of existing high-pressure and normal-pressure models, not only serves to investigate molecular pathogenesis, but also offers more relevant models for testing new therapeutic approaches for multifactorial glaucoma.

## 9. Future Directions

The major challenge in the further development of animal models, especially for NTG, is mimicking the corresponding human conditions as closely as possible. This is quite challenging due to the complexity of the disease. In addition, the “transferability” of these animal models and the knowledge gained from them about glaucoma in humans represents a major hurdle. Future clinical studies using compounds or biologics that work well in animal models will determine the overall significance of these inducible glaucoma models and will guide the further development of animal models. Preclinical animal models are of indispensable value for the discovery and development of drugs, so the focus should be on the further development and combination of existing models. In addition, ethical and bureaucratic hurdles will make the development of preclinical animal models particularly difficult in the future. Organ cultures represent an interesting approach here, as animals do not have to be bred and killed specifically for this purpose. In order to provide adequate treatment options for glaucoma in the future, especially for patients with NTG, it will be necessary to expand the knowledge based on the existing systems and establish a broad range of models.

## Figures and Tables

**Figure 1 ijms-25-00906-f001:**
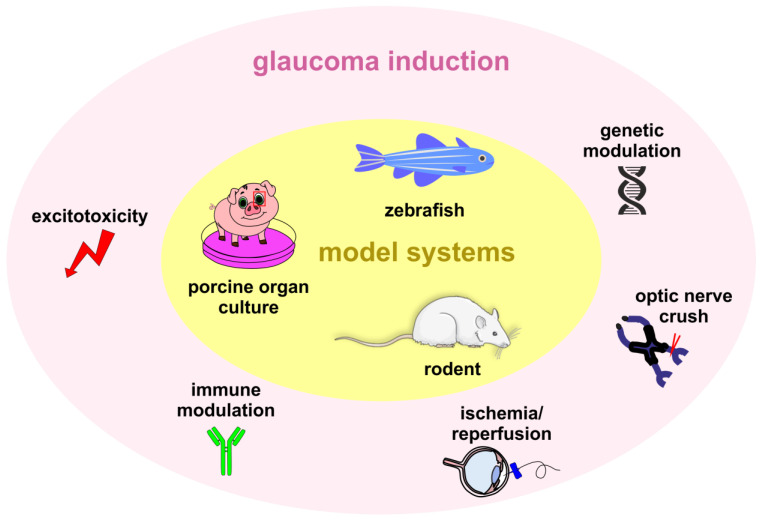
Model systems and induction methods to create glaucoma animal models without chronic IOP elevation. Zebrafish, rodents such as mice and rats, and porcine organ cultures can be used to simulate glaucoma-like damage. To induce glaucoma-like damage in porcine organ cultures of the eye (red box), different excitotoxicity-based substances can be used. In zebrafish, in addition to the excitotoxicity-based substances, genetic modulations can also be carried out to create an NTG model. In rodents, the opportunities range from excitotoxicity-based substances, genetic and immunomodulatory modifications, as well as ischemia/reperfusion induction to optic nerve crush.

**Table 1 ijms-25-00906-t001:** Overview of glaucoma animal models without chronic IOP elevation. Outcomes, advantages, and challenges for these models are listed.

Models		Outcomes	Advantages	Challenges	References
Genetic normal-tension glaucoma models	E50K	Increased apoptosis rate RGC loss Reduction of the retinal thickness	No further intervention necessary	Patients represent approximately 10% of cases worldwide Unclear how this specific degeneration is caused	[50,53,54]
	GLAST, EAAC1	Cupping of the optic nerve head Degeneration of axons GLAST: 8 months EAAC1: 8 weeks	No further interventions necessary	GLAST: long time frame Degeneration is not only focused on a specific area	[69,73,75]
Excitotoxicity-based glaucoma models	NMDA	Rapid degeneration of RGCs after one day Optic nerve degeneration	Usable in ex vivo models	Intravitreal injections	[87,97,101]
Immune-based glaucoma models	EAGs (HSPs, S100B, ONA)	RGC loss Caspase-mediated apoptosis Degeneration of optic nerve Microglia activation	To analyze immune system as an important risk factor	Immunization of animals	[115,118,121]
Ischemia models	Increased IOP, chronic carotid occlusion, transient obstruction of retinal/cerebral artery, vasoconstriction of retinal vessels	Oxidative stress (ROS) Severe neuronal damage (retina and optic nerve) Impaired retinal function Gliosis	Reproducibility Possibility of longitudinal follow-up (OCT, ERG)	Surgical intervention Surgical skills Long anesthesia time	[21,138,147]
Optic nerve crush or transection models		Retrograde degeneration of RGCs Comparable death of RGCs and axons Majority of RGCs die by apoptosis	Reproducibility Labeling of regenerating axons	Surgical intervention Surgical skills	[160,163,168]

## Data Availability

The datasets generated during and/or analyzed during the current study are available from the corresponding author on reasonable request.

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
