# Peer review of "Glaucoma Animal Models beyond Chronic IOP Increase"

_ijms, 2024, doi:10.3390/ijms25020906_

Round 1

Reviewer 1 Report

Comments and Suggestions for Authors

In this review, the investigators stated that they aimed to provide an overview of animal models of glaucoma beyond IOP elevation. In this review, researchers defined glaucoma and provided detailed information about the types and causes of glaucoma. They also provided information about animal models of glaucoma and the roles of these models in the causes and treatment of glaucoma. In the review, the researchers noted that animal models of glaucoma are widely used in research to better understand diseases, discover underlying mechanisms and test potential treatments. In the review, the researchers presented detailed and up-to-date information about glaucoma under 8 main headings with 1 picture and 1 table.

In conclusion, the investigators stated that the mechanisms leading to glaucoma, which is one of the most common causes of blindness, are still not fully understood and noted that other mechanisms play an important role in the development and progression of glaucoma besides IOP increase. In the review, the investigators stated that excitotoxicity, immunological changes, ischaemia and/or oxidative stress contribute to neurodegeneration in glaucoma diseases.

In future directions section, the investigators suggested that in the future it will be necessary to expand knowledge based on existing systems and build a wide range of models to provide adequate treatment options for glaucoma, especially for patients with NTG.

Author Response

See document.

Reviewer 2 Report

Comments and Suggestions for Authors

Reviewer Comments (ijms-2800005-peer-review-v1)

The Review was overviewing the discovered novel therapeutic approaches, appropriate animal models are indeed from glaucoma. It was innovative in some ways, but there was a major concern to be needed to discuss elaborately. I would like the authors to revise it. According to my knowledge, I regret this review manuscript from the IJMS.

1.      Overall, the present review manuscript just listing like essay type, missing the more sub-listing and major discovery with diagrammatic representations.

Need to put sub-title from each heading 1, 2, 3 and 4.

2.      Topic-2 need to be divide sub-title “Types of glaucoma models” with discussing the glaucoma diagnostic markers.

3.      Similarly, topic-2 need to divide two sub-titles like glaucoma, in vitro and in vivo signaling antioxidant pathway in diagrammatic representations.

4.      Topic-4 need to sub-divide and diagrammatic representation of IOP dependent and independent in vivo models.

5.      Similarly, topic-5 to be revise very carefully and need to include the molecular mechanism with diagrammatic representation.

Comments on the Quality of English Language

Nill

Author Response

See document.

Reviewer 3 Report

Comments and Suggestions for Authors

The manuscript by Tsai et al. is an important and timely review that provides a comprehensive update on the current status of animal models of glaucoma, specifically those without chronic IOP elevation. The review is well-balanced and well-written, presenting both historical and recent primary publications and reviews. The authors may want to consider/address the following suggestions/points. 

1.     Lines 47-59 and 72-91 require references.

2.     The spelling of model systems in Fig 1 needs to be corrected.

3.     Consider adding a paragraph about naturally occurring normal tension glaucoma in aged marmosets (PMID: 31960810).

Author Response

See document.

Reviewer 4 Report

Comments and Suggestions for Authors

In my opinion this article has an overview character. This work present that excitoxicity, immunological changes, ischemia, oxidative stres play a major role in neurodegenration in glaucoma. In summary all animal models confirm that they have advantages and disadvantages but play a crucial role in therapeuthic approaches. The topic of the article is original and its address to the researchers in glaucoma and to the pathomorphologists. Very interesting part of this article is ,, Future directions'' which shows that organ cultures can be a new approach used in animal models.  In my opinion references should not be older than 22 years.  

Author Response

See document.

Round 2

Reviewer 2 Report

Comments and Suggestions for Authors

1. Abstract line-24 "Lastly, we discuss optic nerve crush models." The sentence is incomplete and needs revising.

2. Section-2.1. On lines 133 and 134, the author writes "one study", but their references indicate two [46, 47]. Need to revise.

3. The overall outline of the molecular mechanism of signaling pathways with diagrammatic representation is missing in sections 2 to 6.